# Accelerated discovery of multi-elemental reverse water-gas shift catalysts using extrapolative machine learning approach

Gang Wang[1,7], Shinya Mine [1,7], Duotian Chen[1,7], Yuan Jing[1], Kah Wei Ting[1], Taichi Yamaguchi[1], Motoshi Takao[1], Zen Maeno[2], Ichigaku Takigawa [3,4,5] ✉, Koichi Matsushita[6], Ken-ichi Shimizu [1] ✉ & Takashi Toyao [1] ✉

Designing novel catalysts is key to solving many energy and environmental challenges. Despite the promise that data science approaches, including machine learning (ML), can accelerate the development of catalysts, truly novel catalysts have rarely been discovered through ML approaches because of one of its most common limitations and criticisms—the assumed inability to extrapolate and identify extraordinary materials. Herein, we demonstrate an extrapolative ML approach to develop new multi-elemental reverse water-gas shift catalysts. Using 45 catalysts as the initial data points and performing 44 cycles of the closed loop discovery system (ML prediction + experiment), we experimentally tested a total of 300 catalysts and identified more than 100 catalysts with superior activity compared to those of the previously reported high-performance catalysts. The composition of the optimal catalyst discovered was $Pt(3)/Rb(1)\text{-}Ba(1)\text{-}Mo(0.6)\text{-}Nb(0.2)/TiO_2$. Notably, niobium (Nb) was not included in the original dataset, and the catalyst composition identified was not predictable even by human experts.

The discovery of novel catalysts is essential for accelerating the transition to a sustainable future[1,2]. Despite the significant progress in the development of highly efficient catalysts, heterogeneous catalysis remains largely an empirical science owing to the complexity of the underlying surface chemistry[3,4]. Currently, there is a lack of data and design guidelines for heterogeneous catalysis because the computational cost of obtaining accurate theoretical models for such complex systems is currently prohibitively high while high-throughput experimental methods that have been applied successfully in related fields have not yet been thoroughly explored[5–8]. Most of the important catalysts were discovered by chance or through trial-and-error processes

extending over several years; the discovery of truly novel catalysts is still challenging[9].

The recent revolution in data science is expected to accelerate the development of new catalysts significantly, and hence, impact catalysis research[10–14]. Machine learning (ML) will play a central role in this paradigm shift. The application of ML-based approaches to catalysis[15–21] and broader fields of chemistry and materials science has attracted considerable attention[22–27]. Although proof-of-concept examples of reduction in time and cost of catalyst development have been demonstrated using ML-based approaches, most of the ML-based research is directed toward the resolution of benchmark

[1]Institute for Catalysis, Hokkaido University, N-21, W-10, Sapporo 001-0021, Japan. [2]School of Advanced Engineering, Kogakuin University, 2665-1, Nakano-cho, Hachioji 192-0015, Japan. [3]RIKEN Center for Advanced Intelligence Project, 1-4-1 Nihonbashi, Chuo-ku, Tokyo 103-0027, Japan. [4]Institute for Chemical Reaction Design and Discovery (WPI-ICReDD), Hokkaido University, N-21, W-10, Sapporo 001-0021, Japan. [5]Institute for Liberal Arts and Sciences, Kyoto University, 69-302, Yoshida-Konoe-cho, Sakyo-ku, Kyoto 606-8315, Japan. [6]Central Technical Research Laboratory, ENEOS Corporation, 8, Chidori-cho, Naka-ku, Yokohama 231-0815, Japan. [7]These authors contributed equally: Gang Wang, Shinya Mine, Duotian Chen. ✉e-mail: takigawa.ichigaku.8s@kyoto-u.ac.jp; kshimizu@cat.hokudai.ac.jp; toyao@cat.hokudai.ac.jp

problems, while truly novel compounds and materials have rarely been discovered[28,29]. This is due to one of the most common limitations of ML—the assumed inability of the models to extrapolate and identify extraordinary materials beyond those present in the training dataset[30]. In materials and catalysis informatics, we often desire to use ML models to discover an entirely new class of materials and catalysts with unprecedented combinations of elements. In this context, our group has developed a new ML approach wherein elemental features are used as input representations rather than inputting the catalyst compositions directly[31,32]. Namely, each catalyst is represented as a set of elemental descriptors such as electronegativities and melting points, which are scaled by the element content, followed by aggregation into a single feature vector by a permutation-invariant readout operation (elementwise sort pooling, referred to as sorted weighted elemental descriptor (SWED))[31,32]. This ML method can guide catalyst design and discovery in areas where there is limited overlap of catalyst compositions and even for elements that were previously never included in a given dataset, thereby enabling extrapolative and ambitious prediction beyond the training data. Other studies have also validated the possibility of such extrapolative prediction using relevant feature engineering/selection approaches[33]. Despite the theoretical evidence on the possibilities of finding novel catalysts and exceptional materials through extrapolative prediction, the use of ML to identify truly new and exceptional materials has remained elusive[34].

In this study, we have applied the extrapolative ML approach to develop new multi-elemental catalysts based on supported Pt as an active metal and $TiO_2$ as a support for the low-temperature reverse water-gas shift (RWGS) reaction. This reaction was chosen because its product, CO, is an important intermediate in various well-established catalytic processes for manufacturing value-added chemicals; that is, the RWGS reaction enables highly flexible utilization of $CO_2$[35,36].

## Results

### ML-assisted discovery of RWGS catalysts

We explored M elements of up to five types for $Pt(3)/M_1(X_1)$-$M_2(X_2)$-$M_3(X_3)$-$M_4(X_4)$-$M_5(X_5)/TiO_2$ RWGS catalysts (3 wt% Pt, $TiO_2$ = P25). For M, elements with atomic number 3 (Li) through 83 (Bi), except for Be, B, C, N, O, P, S, As, Se, Tc, Te, Pm, Ta, Hg, Tl, halogens, noble gases, and platinum group metals, were used as catalyst components (50 elements in total). Each M element had a unique loading amount (X) for each catalyst. Thus, the total number of catalyst candidates easily exceeded $10^{11}$ even though only integer values of up to 5 wt% were considered as the loading amount of M ($_{50}C_5 \times 5^5 \approx 800$ billion). We have tested three types of ML approaches, each of which differs in the input representations of the catalysts: (i) a *naive* ML model, which uses only elemental compositions; (ii) an *exploitative* ML model, which uses both elemental compositions and elemental properties; and (iii) an *explorative* ML model, which uses only elemental properties. For the input representation of the elemental compositions, each catalyst was represented as a vector of the compositional fractions for all the 50 elements under consideration. On the other hand, for the input representation of the elemental properties, vectors of 8 selected elemental descriptors for each element, scaled by its composition fraction, were aggregated into a single feature vector by sum pooling. Therefore, the naive, exploitative, and explorative ML models had 50, 58, and 8 descriptor dimensions, respectively. The initial dataset consisting of 45 data points was constructed using the catalysts reported in our previous experimental study[37] and some catalysts fabricated in the present study (See the data directory in the GitHub repository https://github.com/shinya-mine); this dataset was set as "Iteration" = 0. We then trained the explorative ML model based on Extra-Trees regression (ETR)[38] with the initial dataset (45 data points), calculated the expected improvement (EI) for all the test points in the catalyst composition grid, selected several prominent catalyst candidates considering the EI values and catalyst variety, synthesized the

catalysts using the sequential impregnation method, performed the RWGS reaction, and updated the dataset to close the loop (Supplementary Fig. 1). We continued this process for 44 loops to test 300 catalysts, as shown in Fig. 1. The explorative ML model was used in the initial effort to explore many elements, and because the model achieved the highest prediction accuracy among the three ML models. The exploitative ML model was used after the prediction accuracy reached a certain level (after 30 iterations). Although the naive ML model was not used for the catalyst discovery process in this study, its prediction results are given for comparison, because fractional representation in a one-hot encoding manner is known to perform as well as or better than many other featurization techniques when large datasets are used ref. 29.

Through experimental testing of 255 ML-predicted new catalysts corresponding to 44 cycles of the closed loop discovery system (ML prediction + experiment), we found more than 100 catalysts that showed higher activity than the previously reported high-performance catalyst $(Pt(3)/Mo(10)/TiO_2)$[37] (Fig. 1). In the early stages, the prediction accuracy of the ML model was not high; thus, finding good catalysts was difficult. However, as the amount of data increased and the prediction accuracy improved, we were able to identify good catalysts. This is widely known as the exploration–exploitation trade-off in ML, where we need to balance between "exploration" to obtain more data on uncertain parts and "exploitation" to rely on the already obtained data. Comparing the radar charts of the elemental descriptors for the best catalysts at each iteration (Fig. 1B) shows how the properties of each catalyst composition changed with successive iterations. Although our dataset is still small (300 data points) and the best prediction accuracy attained after 44 cycles ($R^2 = 0.81$) is not significantly high, the proposed design is iterative, i.e., a sequential experimental design. Thus, the focus is more on how to utilize the available data (even if the dataset is small in the statistical sense) to plan subsequent experiments and achieve better catalyst discovery. We believe that the prediction accuracies (up to $R^2 = 0.81$) achieved by a standard cross validation (CV) procedure (see the ML method section for details) would be sufficient to statistically sense promising directions for further research. It is also noteworthy that the obtained prediction accuracy ($R^2 = 0.81$) is somewhat higher than those attained in most ML studies using experimental data on heterogeneous catalysis and relevant material science topics, wherein the prediction accuracy is typically below $R^2 = 0.8$, even when experimental conditions are used as descriptors[28,31,32,39–42]. The composition of the best catalyst discovered by this approach was $Pt(3)/Rb(1)$-$Ba(1)$-$Mo(0.6)$-$Nb(0.2)/TiO_2$, and it exhibited the highest CO formation rate per unit catalyst mass (mmol $min^{-1}$ $g_{cat}^{-1}$) at temperatures below 250 °C compared with the previously reported catalysts, while retaining 100% CO selectivity (Supplementary Table 4). Commercial water-gas shift catalysts[43] such as $Cu/ZnO/Al_2O_3$ (HiFUEL® W220) and $FeCrCuO_x$ (HiFUEL® W210) were tested and found to be ineffective in this low temperature range (Supplementary Table 5). Control studies confirmed that all the components are necessary to obtain the highest CO formation rate. All the CO formation rates were tested at least three times, and the average values are shown in Supplementary Fig. 7, along with error bars representing the data range. Notably, Nb was not included in the original dataset (Fig. 2), and the identified catalyst composition could hardly be predicted even by human experts. The compositions of the second, third, and fourth best catalysts are $Pt(3)$-$Mo(0.8)$-$Ba(0.7)$-$Na(0.4)$-$Ce(0.2)/TiO_2$, $Pt(3)/Rb(1)$-$Ba(1)$-$Mo(0.6)$-$Eu(0.4)/TiO_2$, and $Pt(3)/Tb(2)$-$Sm(1.5)$-$Ce(1.2)$-$Re(1.2)$-$Mo(0.6)/TiO_2$, respectively. Note that we tested the performance of these top-four catalysts and the catalysts highlighted in the radar charts in Fig. 1 at least three times, and the reported values are the averages of these tests.

The extrapolative search is driven by our coarse-grained abstraction of the feature representations (i.e., the descriptors of catalysts) rather than the ML model architecture. Typically, each element

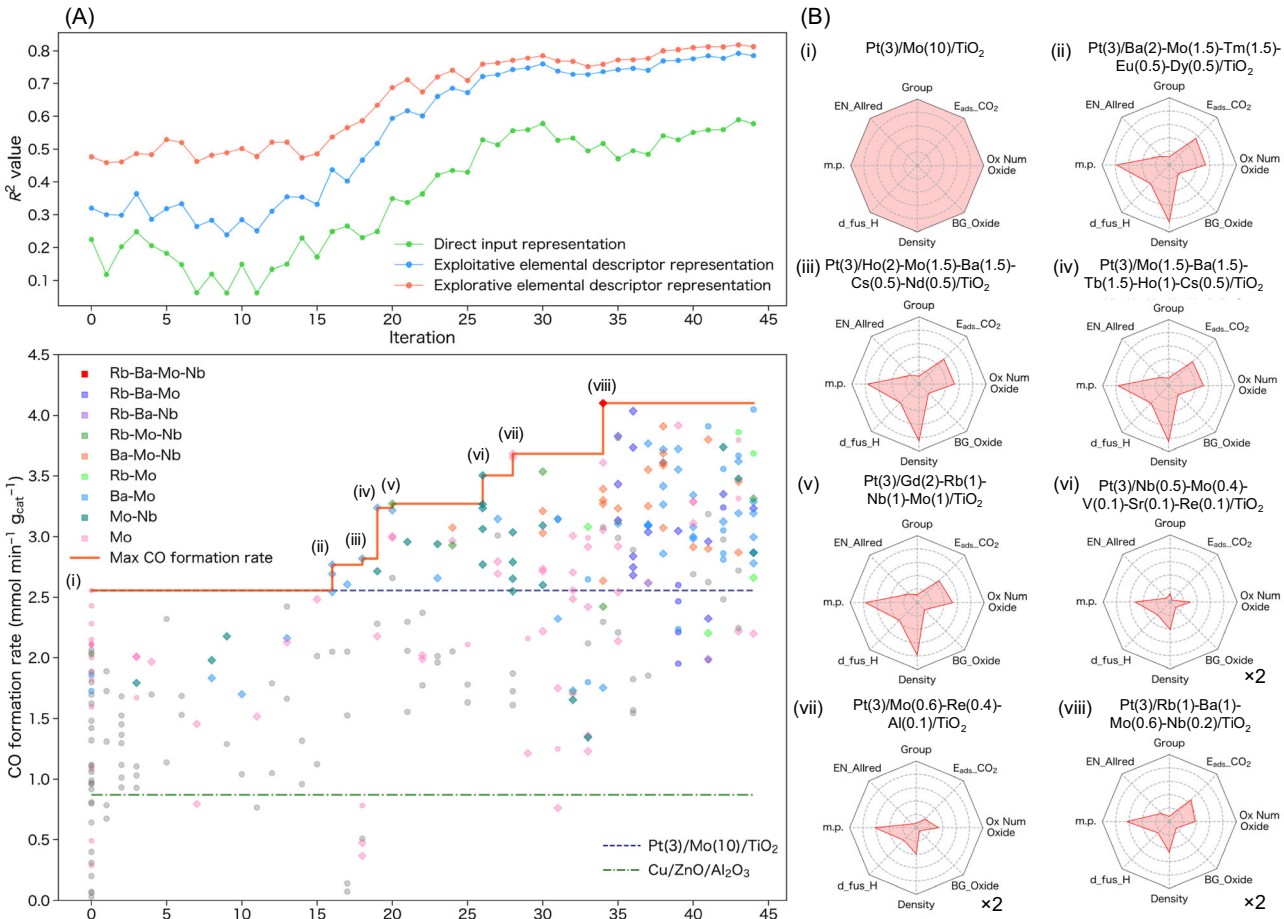

**Fig. 1 | ML-assisted exploration of RWGS catalysts. A** ML-assisted exploration of RWGS catalysts using the explorative and exploitative ML methods based on ETR. Catalysts with elements not present in the original dataset are shown with diamond-shaped symbols while catalysts with elements in the original dataset are shown with gray-colored and circle-shaped symbols. The solid red line shows the best CO formation rate at each iteration, and for comparison, the dashed navy and dash-dotted green lines show the CO formation rates for Pt(3)/Mo(10)/TiO$_2$ and Cu/ZnO/Al$_2$O$_3$ catalysts, respectively. The $R^2$ values were calculated using the cross valida-tion (CV) method described in the ML methods section on the dataset at each iteration before experimental validation. **B** Radar charts of the elemental descrip-tors for the best catalysts at each iteration. Descriptor values relative to the (i) Pt(3)/Mo(10)/TiO$_2$ catalyst are shown.

of a catalyst represents an individual coordinate in a search space; thus, the catalyst composition is represented in a one-hot encoding manner, for example, Mo 10 or Rb 1 Ba 1 Mo 0.6 Nb 0.2. By contrast, we used the feature representations describing each catalyst by elemental descriptors[31,32], i.e., not directly representing elements as distinct symbols but representing them as continuous quantities characterized by a user-chosen set of elemental properties, such as electronegativity and density (as seen in Fig. 1B). We believe that interpolating the tar-geted properties over this abstracted representation can lead to some out-of-training discovery, which we refer to as "extrapolative;" this includes catalysts containing elements never used in the training dataset. In addition, in this study, we used eight descriptors, and the descriptors have eight dimensions, resulting in lower dimensionality than the direct input representation that has 50 dimensions (50 ele-ments). This low dimensionality for the explorative model may have contributed to its success by narrowing the search space.

For ML models, we primarily used tree-ensemble models that are equivalent to a histogram over data-dependent partitions. The tree-ensemble models make conservative predictions in the out-of-training regions (it is a histogram approximation, and any predicted values are the local averages of the training samples, even in the out-of-training regions). In that sense, our approach is based on highly safe/con-servative predictions; nevertheless, it successfully found some cata-lysts containing elements not in the training data, which is worth

emphasizing. Namely, our ML method can extrapolate from the per-spective of materials science as it can identify new elements by moving across the periodic table, while it interpolates from a data science perspective within the elemental descriptor representations. The essential operation of ML prediction is grounded in the interpolation of the given data points; thus, no ML model architecture can directly make extrapolative predictions without further encoding any physics or data-independent hypotheses.

Note that we observe overfitting to the training data and a non-negligible gap between the training and test errors, as shown in Sup-plementary Figs. 11 and 15. This phenomenon, known as "benign/harmless overfitting," is a topic of ongoing discussion in the field of ML[44–46]. In principle, ETR works as a pseudo-piecewise-linear inter-polation, and in cases where the number of data points is limited, interpolating noisy training data can provide more informative pre-dictions than attempting to separate the noise from the data in such underspecified cases with small samples, as shown in Supplemen-tary Fig. 4.

Figure 2A, B shows histograms of the component elements for our dataset which is composed of 300 experimental data with unique catalyst compositions including 50 elements. Elements Mo, Ba, and Nb appeared most frequently. The effect of the loading amount of some of the frequently appearing elements including Mo, Ba, Nb, Re, Rb, and Cs is shown in Supplementary Fig. 9. Catalysts having relatively low

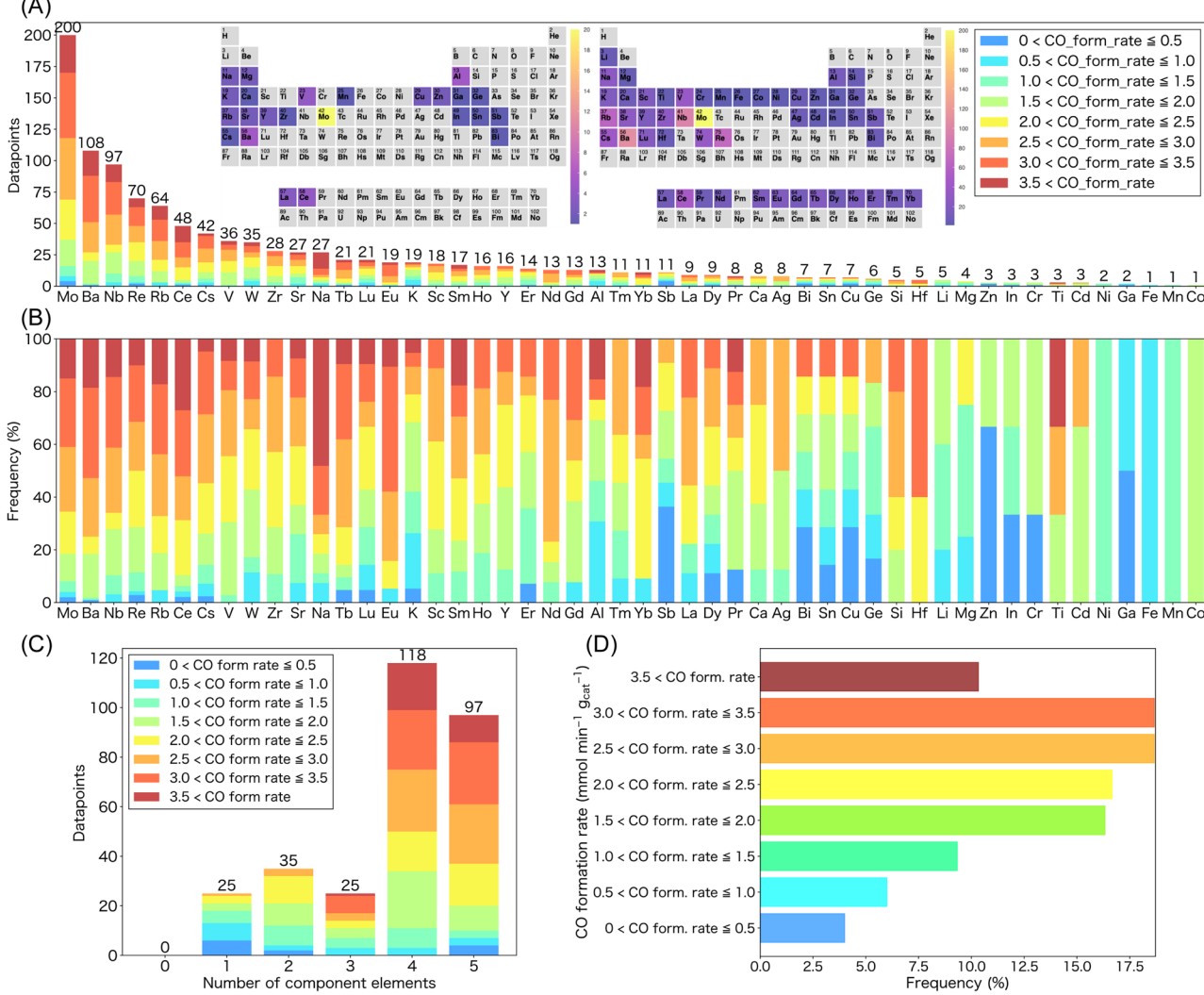

**Fig. 2 | Visualization of RWGS catalyst datasets. A, B** Histograms for each additive oxide component categorized by the RWGS activity; the elements in the original (left) and final (right) datasets are shown in the periodic tables. The maximum values on the Y axis for (**A**) represent the sum of the number of data points while that for (**B**) represent percentage of the RWGS activity category. **C** Number of component elements as additive oxides. **D** Frequency of RWGS catalysts showing different activities.

loading amounts of additive oxides (below 2 wt%) tend to show high CO formation rates.

## Statistical analysis using ML

Although ML is often employed as a black box without any prior insight into what the model has actually learned, supervised ML models can be used to identify important chemical moieties influencing the prediction, even without any explicit knowledge of its underlying principles[47]. Extrapolative ML can reveal not only the effective catalyst compositions but also the required elemental features and electronic properties for the precise designing of ideal catalysts. Feature-importance score and SHapley Additive exPlanations (SHAP)[48,49] analyses were used to understand the importance of the descriptors for ML prediction, as shown in Fig. 3A, B, respectively. Elemental properties such as group, electronegativity (EN), and density were identified as important factors. SHAP can be used to visualize the dependence of the model output (e.g. CO formation rate) on the value of each descriptor[31]. For example, relatively low values (red color in Fig. 3B) for the feature "group" are correlated to a high CO formation rate (SHAP value). The feature-importance score and SHAP analyses were also performed using the exploitative elemental descriptor representation because this method considers the elemental composition directly and

facilitates the understanding of the contribution of the elements in the given data (Supplementary Fig. 16). For the catalyst composition, Mo, Tb, Na, and Ba were identified as important descriptors. The SHAP values were analyzed using waterfall plots for the two representative catalysts (Pt(3)/Rb(1)-Ba(1)-Mo(0.6)-Nb(0.2)/TiO₂ and Pt(3)/Mo(10)/TiO₂), as shown in Fig. 3C, D. The waterfall plot analysis reveals the descriptors that are responsible for the increase or decrease from the average value of the dataset (2.28) relative to the predicted value for each catalyst. EN, group, and oxide band gap (BG) values were found to strongly contribute to the high activity of our best catalyst (Pt(3)/Rb(1)-Ba(1)-Mo(0.6)-Nb(0.2)/TiO₂). Note that the summary plot shown in Fig. 3B describes overall predictions for all the datapoints used (300 datapoints here) whereas the waterfall plots (Fig. 3C, D) are designed to display explanations for individual predictions for each catalyst[48,49]. This difference in methodology is reflected in the differences in ranking of important descriptors in each analysis method. Therefore, the summary plot is useful for obtaining information on the catalyst design guidelines for the RWGS reaction in general, whereas the waterfall plots provide more useful information on the reasons for the high (or low) activity shown by an individual catalyst. The waterfall plots for some additional catalysts are also included in Supplementary Figs. 13, 14, 17 and 18.

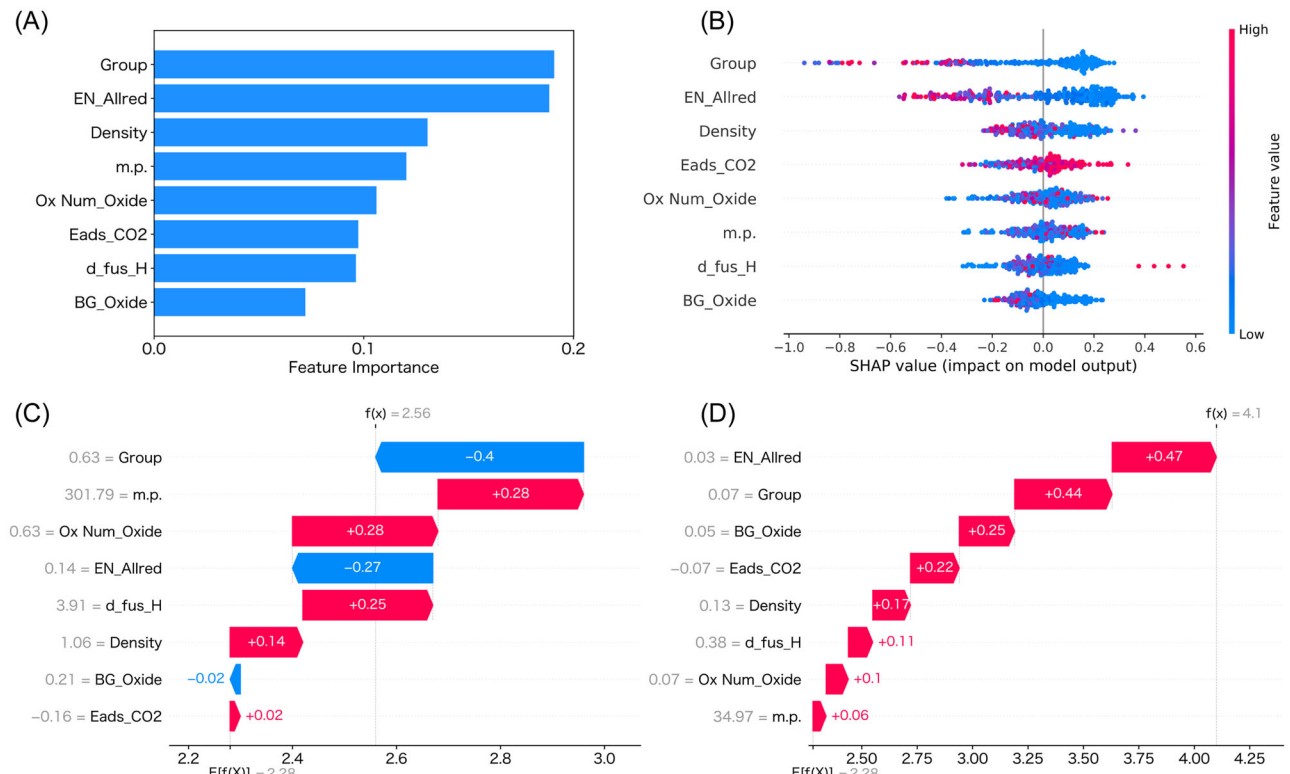

**Fig. 3 | ML-assisted statistical analysis. A** Feature-importance scores and (**B**) SHAP values of the descriptors (summary plot) used to predict CO formation rates of all the 300 catalysts in our final dataset (red and blue for SHAP analysis correspond to high and low features, respectively). Features are in the descending order of the sum of their absolute SHAP values. Dots are displaced vertically to reflect the density of data points at a given SHAP value. Breakdown of SHAP values as waterfall plots for (**C**) the original best catalyst Pt(3)/Mo(10)/TiO$_2$ and (**D**) the current best Pt(3)/Rb(1)-Ba(1)-Mo(0.6)-Nb(0.2)/TiO$_2$ to determine the feature values that are responsible for the increase or decrease from the base. Positive and negative contributions of each feature are shown in red and blue, respectively. Explorative elemental descriptor representation was used.

## Catalyst characterization

With the best catalyst composition in hand, we then performed structural analysis (Fig. 4, Supplementary Figs. 19–27, Supplementary Tables 6 and 7) and mechanistic studies (Fig. 5, Table 1, and Supplementary Figs. 28–33). This is important because investigations of extraordinary materials can provide new scientific insights. The X-ray diffraction pattern of Pt(3)/Rb(1)-Ba(1)-Mo(0.6)-Nb(0.2)/TiO$_2$ was essentially the same as that of pristine TiO$_2$ (P25) and showed peaks corresponding to both anatase and rutile phases (Supplementary Fig. 19). To investigate the morphologies and particle sizes of the introduced Mo and Pt species, high-angle annular dark-field scanning transmission electron microscopy (HAADF-STEM) was performed for TiO$_2$ (P25), Rb(1)-Ba(1)-Mo(0.6)-Nb(0.2)/TiO$_2$, and Pt(3)/Rb(1)-Ba(1)-Mo(0.6)-Nb(0.2)/TiO$_2$ (Fig. 4A). The oxide additive species was found to be highly dispersed over the TiO$_2$ surface. In addition, the Pt nanoparticles in Pt(3)/Rb(1)-Ba(1)-Mo(0.6)-Nb(0.2)/TiO$_2$ were highly dispersed, with an average Pt particle diameter of 1.8 nm (Supplementary Fig. 22). Comparison with the previously identified Pt(3)/Mo(10)/TiO$_2$ active catalyst (particle size of 2.6 nm)[37] revealed that the average particle size of the supported Pt was smaller in Pt(3)/Rb(1)-Ba(1)-Mo(0.6)-Nb(0.2)/TiO$_2$.

X-ray absorption spectroscopy (XAS) was conducted to identify the chemical states of the introduced species in the RWGS catalyst (Fig. 3B and Supplementary Fig. 24). The Pt L$_3$-edge X-ray absorption near-edge structure (XANES) of the reduced Pt(3)/Rb(1)-Ba(1)-Mo(0.6)-Nb(0.2)/TiO$_2$ catalyst was identical to that of the Pt foil used as the reference. Extended X-ray absorption fine structure analysis shows the presence of Pt–Pt bond with coordination number of 5.6 at 2.75 Å (Supplementary Table 7). The observed distance is slightly shorter than that of the Pt–Pt bond observed in bulk Pt metal (2.76 Å),

revealing the formation of nanoparticles[50] that were also found by STEM. Mo K-edge XANES showed that the shape and edge position of the unreduced Pt(3)/Rb(1)-Ba(1)-Mo(0.6)-Nb(0.2)/TiO$_2$ catalyst were identical to those of the reference MoO$_3$. For the reduced Pt(3)/Rb(1)-Ba(1)-Mo(0.6)-Nb(0.2)/TiO$_2$ sample, the absorption edge shifted toward lower energies, indicating the reduction of the Mo species upon pretreatment with H$_2$. X-ray photoelectron spectroscopy (XPS) measurements were conducted to identify the oxidation states of Mo (Fig. 4C). Peaks corresponding to Mo$^{4+}$ were predominantly observed, in addition to small peaks of Mo$^{6+}$ and Mo$^{2+}$. The other additives, including Rb, Ba, and Nb, did not change their oxidation states and existed in the form of Rb$_2$O, BaO, and Nb$_2$O$_5$, respectively, after the reduction pretreatment with H$_2$ (Supplementary Fig. 26).

In situ CO adsorption IR spectroscopy experiments were conducted to examine the electronic state of the Pt species on a series of supported Pt catalysts to understand the effects of the introduced additives (Fig. 4D). All the spectra showed a peak at 2071–2077 cm$^{-1}$, corresponding to the CO bound to the on-top sites of the metallic Pt surface. The center of the CO adsorption peak shifted to higher wavenumbers, following the order Pt(3)/TiO$_2$, Pt(3)/Rb(1)-Ba(1)-Mo(0.6)/TiO$_2$, Pt(3)/Mo(0.6)/TiO$_2$ and Pt(3)/Rb(1)-Ba(1)-Mo(0.6)-Nb(0.2)/TiO$_2$. Therefore, the introduction of additives favors the formation of more electron-deficient metallic Pt$^0$ species, compared to pristine Pt(3)/TiO$_2$, and is expected to improve the resistance to CO poisoning during the RWGS reaction. The same trend was also observed by XPS (Supplementary Fig. 27).

## Mechanistic studies

Kinetic studies were conducted on the optimal catalyst (Pt(3)/Rb(1)-Ba(1)-Mo(0.6)-Nb(0.2)/TiO$_2$). The apparent activation energy ($E_a$), as

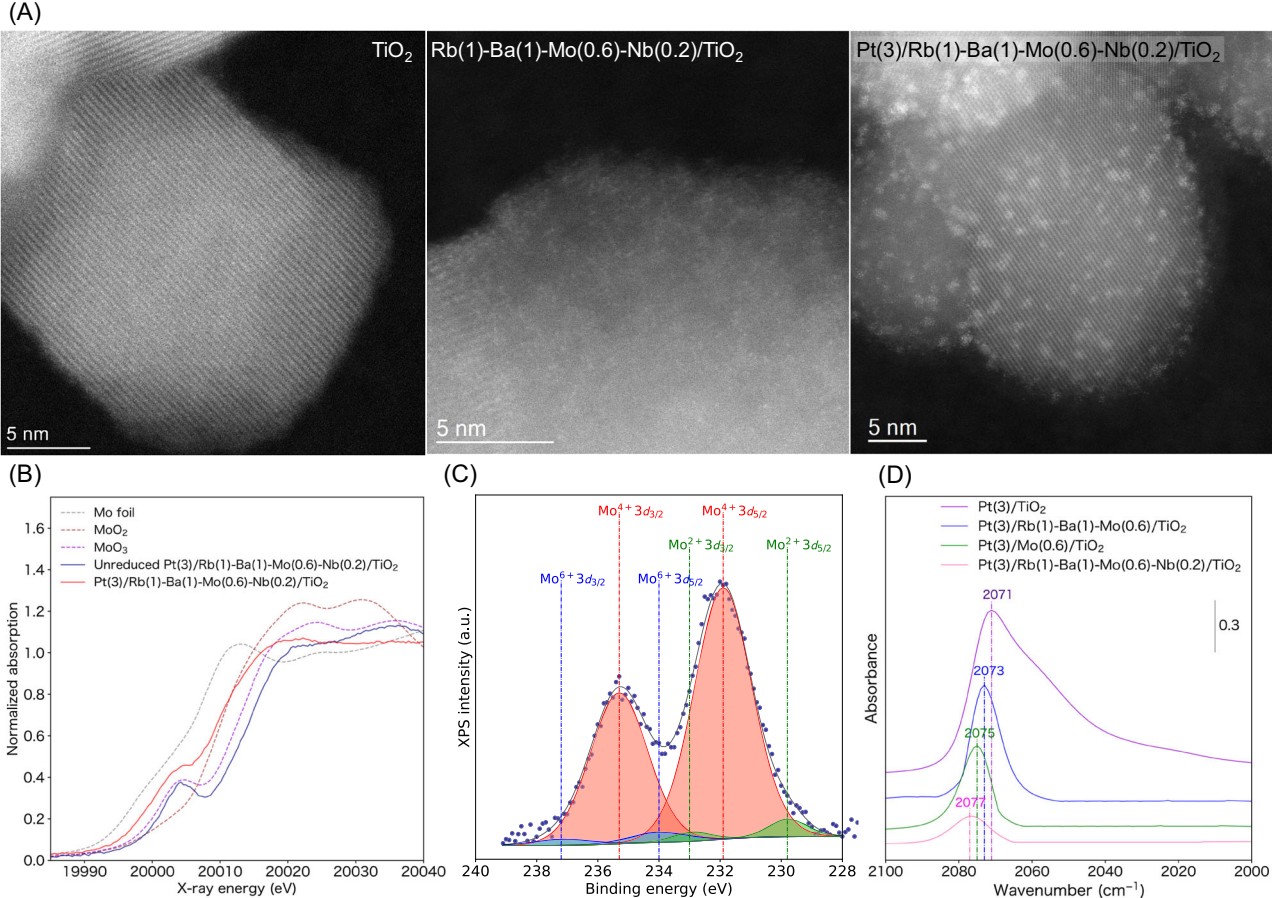

**Fig. 4 | Structural analysis of the ML-identified RWGS catalyst. A** HAADF-STEM images of $TiO_2$, Rb(1)-Ba(1)-Mo(0.6)-Nb(0.2)/$TiO_2$, and Pt(3)/Rb(1)-Ba(1)-Mo(0.6)-Nb(0.2)/$TiO_2$. **B** Mo K-edge XANES of unreduced and reduced Pt(3)/Rb(1)-Ba(1)-Mo(0.6)-Nb(0.2)/$TiO_2$ and reference compounds. **C** XPS spectra of the supported Pt catalysts after the $H_2$ reduction pretreatment at 300 °C without exposure to air. **D** IR spectra of CO adsorbed on the supported Pt catalysts, recorded at 250 °C after the $H_2$ reduction pretreatment at 300 °C. The sample was exposed to a flow of 1% CO/He (100 mL min$^{-1}$) for 5 min and purged with He for 5 min.

calculated from the Arrhenius plot, was 45.6 kJ mol$^{-1}$ (Table 1 and Supplementary Fig. 28). Similarly, the $E_a$ values of Pt(3)/Rb(1)-Ba(1)-Mo(0.6)/$TiO_2$, Pt(3)/Mo(0.6)/$TiO_2$, and Pt(3)/$TiO_2$ were 48.7, 52.8, and 58.4 kJ mol$^{-1}$, respectively. The apparent reaction orders with respect to $H_2$, $CO_2$, and CO were calculated to understand the effect of the introduced additives. The apparent reaction orders for both $CO_2$ and $H_2$ in the case of the catalyst with oxide additives decreased as compared with those for pristine Pt(3)/$TiO_2$, indicating weaker dependence on their concentrations. In addition, the reaction order with respect to CO was the smallest for Pt(3)/Rb(1)-Ba(1)-Mo(0.6)/$TiO_2$, indicating less inhibitory effect of CO for the best catalyst. This result is consistent with the results of the in situ IR and XPS experiments. These combined results indicate that the introduction of Nb renders Pt more electron-deficient and induces high tolerance to CO poisoning, leading to a high catalytic activity. The $CO_2$-TPD analysis of the catalysts without Pt (Supplementary Fig. 29) suggested that the introduced additives could facilitate the adsorption of $CO_2$ owing to the introduced base metal oxides, particularly Rb and Ba, thereby promoting the reaction efficiently.

The RWGS reaction is known to proceed mainly via the (i) redox mechanism and (ii) associative mechanism[51]. In the former, oxygen vacancies are formed on the surface of the support oxide by $H_2$, while $CO_2$ reoxidizes the partially reduced oxide to fill the formed oxygen vacancies[52], resulting in the formation of CO. In the latter mechanism, CO is produced through the decomposition of the surface-reactive intermediates such as formates and carbonates[51].

To elucidate the reaction mechanism, *operando* XANES measurements were conducted under $CO_2$, $H_2$, and $CO_2$ + $H_2$ flow at 250 °C (Fig. 5). The Mo K-edge XANES spectra of Pt(3)/Rb(1)-Ba(1)-Mo(0.6)-Nb(0.2)/$TiO_2$ show that the absorption edge shifts to higher energies after the introduction of $CO_2$, while CO was simultaneously detected by GC. The results clearly demonstrated that $CO_2$ acted as an oxidant to oxidize the Mo species. Notably, CO was formed even upon the introduction of $H_2$, suggesting that the reaction also proceeded through the associative mechanism. For the Pt $L_3$-edge (Supplementary Fig. 30), the white line intensity became slightly stronger under $CO_2$ flow, suggesting that metallic Pt was also oxidized by $CO_2$. Note that this change can be solely because of the adsorption of the CO formed, as it is well-known that the Pt $L_3$-edge XANES intensity and shape is altered by the adsorption of CO[53]. The K-edge XANES spectra of Ti, Ba, Rb, and Nb were also obtained employing a protocol similar to that described above (Supplementary Fig. 30). The edge positions in all these XANES spectra hardly changed following the introduction of $CO_2$, indicating that no redox reactions of $TiO_2$, BaO, $Rb_2O$, and $Nb_2O_5$ occurred during the RWGS reaction.

*Operando* IR spectroscopy was also performed to investigate the adsorbed surface species that are likely to be involved in the RWGS reaction (Fig. 5B). Bands in the range 1700–1200 cm$^{-1}$, which can be assigned to the surface-adsorbed species such as carbonate and formate[51], appeared immediately after the introduction of $CO_2$. Simultaneous formation of CO in the gas phase was also observed using an IR gas cell at the outlet. Bands at 2100–1950 cm$^{-1}$, which can be assigned to the adsorbed CO, were also observed. The amount of

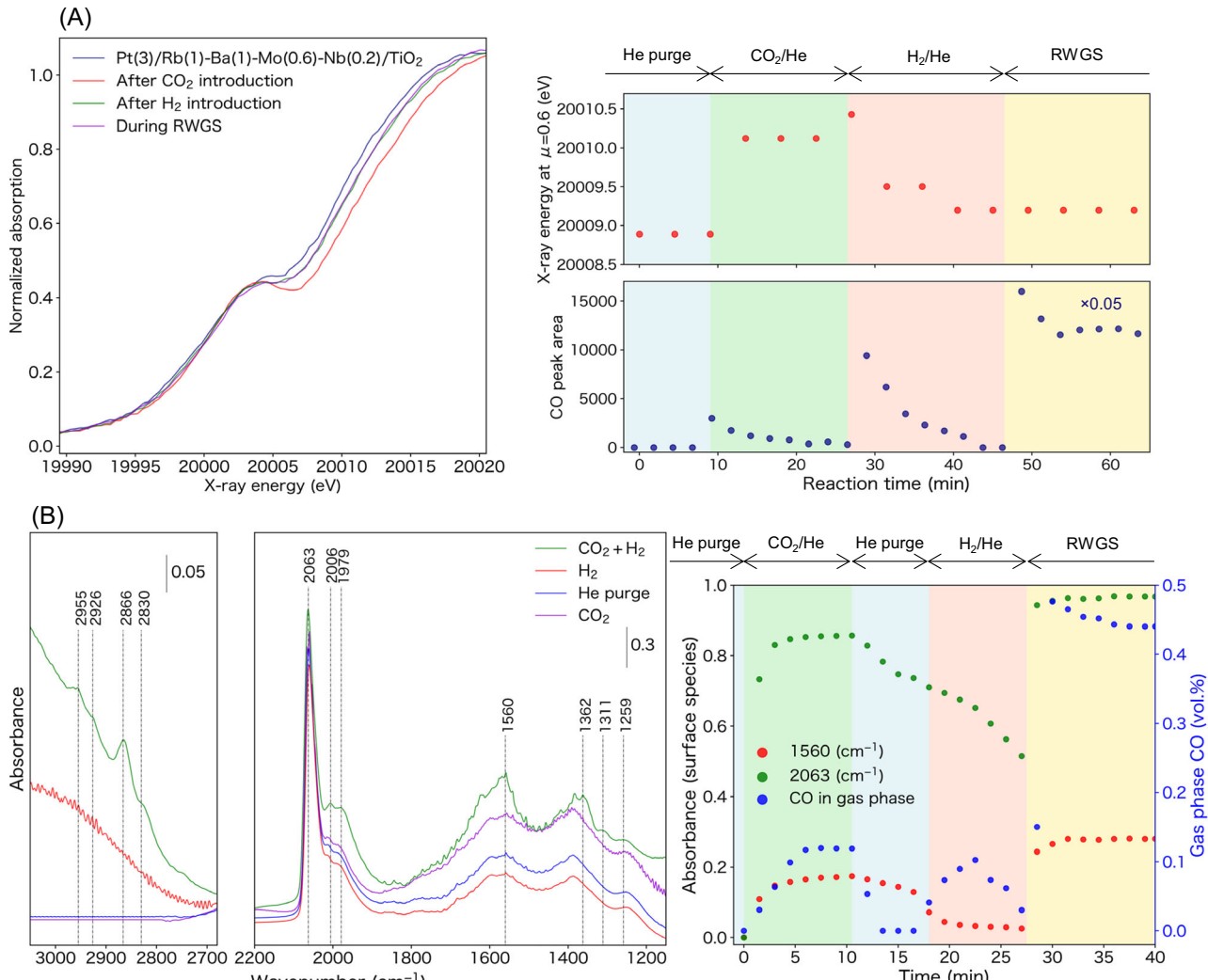

**Fig. 5 | *Operando* spectroscopic studies. A** *Operando* Mo K-edge XANES spectra of Pt(3)/Rb(1)-Ba(1)-Mo(0.6)-Nb(0.2)/TiO$_2$ obtained under a sequential flow of 25% CO$_2$/He, 75% H$_2$/He, and 25% CO$_2$ + 75% H$_2$ at 250 °C (left). Changes in the X-ray energy (at $\mu$ = 0.6 eV) and CO concentration in the gas phase (right). **B** *Operando* IR

measurements for the Pt(3)/Rb(1)-Ba(1)-Mo(0.6)-Nb(0.2)/TiO$_2$ catalyst conducted under a sequential flow of CO$_2$, He, H$_2$, and 25% CO$_2$ + 75% H$_2$ at 200 °C (left). Variations in the intensities of the peaks related to the surface-adsorbed species and concentration of CO in the effluent gas upon the introduction of CO$_2$ (right).

these surface species over the best catalyst was higher than those over Pt(3)/Mo(0.6)/TiO$_2$ and Pt(3)/TiO$_2$, yet lower than that over Pt(3)/Rb(1)-Ba(1)-Mo(0.6)/TiO$_2$ without Nb (Supplementary Fig. 31). The evolution of the bands in the $\nu_{CH}$ region (2800–2960 cm$^{-1}$) also supports the formation of formate species under the flow of CO$_2$ and H$_2$.

**Table 1 | Apparent reaction orders and activation energy ($E_a$) for the RWGS reaction over Pt(3)/Rb(1)-Ba(1)-Mo(0.6)-Nb(0.2)/TiO$_2$, Pt(3)/Rb(1)-Ba(1)-Mo(0.6)/TiO$_2$, Pt(3)/Mo(0.6)/TiO$_2$, and Pt(3)/TiO$_2$ catalyst**

| Catalyst | CO$_2$ [a] | H$_2$ [b] | CO [c] | $E_a$ (kJ mol$^{-1}$) |
|---|---|---|---|---|
| Pt(3)/Rb(1)-Ba(1)-Mo(0.6)-Nb(0.2)/TiO$_2$ | 0.47 | 0.48 | -0.80 | 45.6 |
| Pt(3)/Rb(1)-Ba(1)-Mo(0.6)/TiO$_2$ | 0.39 | 0.42 | −1.12 | 48.7 |
| Pt(3)/Mo(0.6)/TiO$_2$ | 0.48 | 0.52 | −1.01 | 52.8 |
| Pt(3)/TiO$_2$ | 0.53 | 0.57 | −1.37 | 58.4 |

[a]Catalyst (10 mg), 0.706 atm H$_2$, total flow rate of CO$_2$, H$_2$ and N$_2$ is 85 mL min$^{-1}$, 250 °C.
[b]Catalyst (10 mg), 0.235 atm CO$_2$, total flow rate of CO$_2$, H$_2$ and N$_2$ is 85 mL min$^{-1}$, 250 °C.
[c]Catalyst (10 mg), CO$_2$/H$_2$ = 1/3, total flow rate of CO$_2$, H$_2$, N$_2$ and CO (1.0-2.5 mL min$^{-1}$) is 85 mL min$^{-1}$, 250 °C.

These results indicate that the Ba and Rb species act as base components to generate the surface-adsorbed species that lead to the formation of CO. To confirm this, H$_2$ was introduced to the Pt(3)/Rb(1)-Ba(1)-Mo(0.6)-Nb(0.2)/TiO$_2$ catalyst with such adsorbed species, as shown in Fig. 5B and Supplementary Fig. 33. Note that for this purpose, a lower temperature (200 °C) was employed to clearly observe the adsorbate peaks. Intensities of the bands between 1700 and 1200 cm$^{-1}$ decreased upon the introduction of H$_2$, and simultaneous formation of CO in the gas phase was observed. These *operando* XAS and IR results indicated that Mo acted as a redox species while Rb and Ba acted as bases to promote the RWGS reaction. Nb was not directly involved in the reaction; it rather modified the electronic structure of Pt, ensuring high CO tolerance. These multiple functions realized by the combination of the oxide additives identified are vital for achieving high catalytic performance.

**Catalyst durability**
Finally, a durability test was conducted (Fig. 6). For the optimal Pt(3)/Rb(1)-Ba(1)-Mo(0.6)-Nb(0.2)/TiO$_2$ catalyst, the CO yield after 1 h time-on-stream was observed as 8.0% with the corresponding CO formation rate of 3.34 mmol min$^{-1}$ g$^{-1}$. Note that 100% CO selectivity was retained throughout the durability test. Although the CO yield decreased

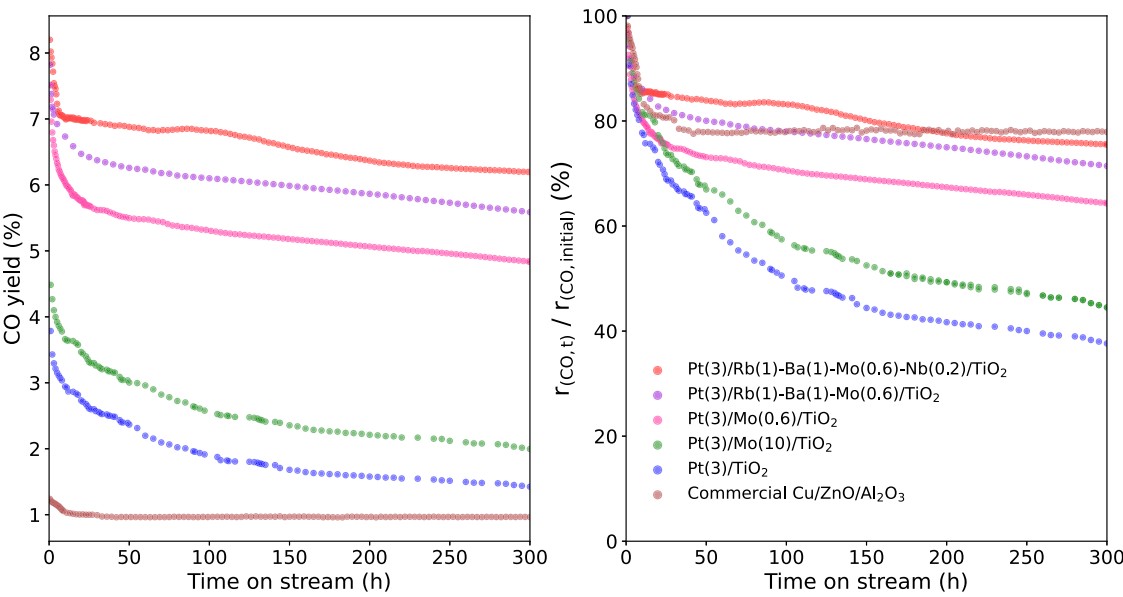

**Fig. 6 | Long-term stability.** Durability test for the supported Pt and commercial Cu/ZnO/Al$_2$O$_3$ catalysts at the RWGS reaction conditions of 10 mg catalyst, 10 mL min$^{-1}$ CO$_2$, 30 mL min$^{-1}$ H$_2$, and 5 mL min$^{-1}$ N$_2$ (internal standard for GC analysis), 250 °C and 1 atm.

gradually over time, the CO formation rate after 300 h time-on-stream was still 2.52 mmol min$^{-1}$ g$^{-1}$. For comparison, the catalytic stabilities of Pt(3)/Rb(1)-Ba(1)-Mo(0.6)/TiO$_2$, Pt(3)/Mo(0.6)/TiO$_2$, Pt(3)/TiO$_2$, Pt(3)/Mo(10)/TiO$_2$ (reported previously by our group)[37] and a commercial Cu/ZnO/Al$_2$O$_3$ catalyst were also evaluated under the same reaction conditions. The CO yields obtained over these reference supported Pt catalysts were all lower than that on Pt(3)/Rb(1)-Ba(1)-Mo(0.6)-Nb(0.2)/TiO$_2$ throughout the durability test time period. Although the Cu/ZnO/Al$_2$O$_3$ catalyst exhibited relatively good stability for RWGS reaction under our conditions, its activity is much lower than that of the supported Pt catalysts. We also compared the degree of the activity loss for each catalyst ($r_{CO, t}/r_{CO,initial}$). It is observed that the optimal Pt(3)/Rb(1)-Ba(1)-Mo(0.6)-Nb(0.2)/TiO$_2$ is comparable to Cu/ZnO/Al$_2$O$_3$ even for this criterion. Therefore, the optimal Pt(3)/Rb(1)-Ba(1)-Mo(0.6)-Nb(0.2)/TiO$_2$ predicted by ML method is an outstanding state-of-the-art catalyst for the low-temperature (250 °C) RWGS reaction.

## Discussion

In summary, using the extrapolative ML method, we discovered over 100 catalysts that produced higher activity than the previously reported best catalyst (Pt(3)/Mo(10)/TiO$_2$). The composition of the optimal discovered catalyst was Pt(3)/Rb(1)-Ba(1)-Mo(0.6)-Nb(0.2)/TiO$_2$. This unique composition could not be predicted by human experts in catalysis; therefore, computational methods, such as ML, would be required to design effective catalysts. Notably, Nb was absent in the original dataset, highlighting the effectiveness of our extrapolative ML model. We also used ML analysis to identify the physical and chemical properties that governed the catalytic activity. Our ML model revealed the effective catalyst compositions as well as the elemental features and electronic properties required for catalytic activity. Experimental mechanistic studies using in situ/*operando* techniques were also performed to explore the role of each catalyst component and the reaction mechanism. The obtained results indicated that Mo acted as a redox species, whereas Rb and Ba acted as bases to promote the RWGS reaction. By contrast, Nb did not directly participate in the reaction but instead altered the electronic structure of Pt, increasing the CO tolerance. Our study presents a new approach for discovering novel catalysts and materials that show extraordinary performance. Although we focused on investigating the effect of the catalyst composition only on the catalytic performance to limit the

search space without changing the experimental conditions, we are aware that the preparation processes can significantly influence the structure of catalysts, which, in turn, can result in variations in the catalytic performance. Further studies are needed to explore the effect of altering the experimental conditions by using ML, even though that will necessitate a considerably large number of experiments. In addition, full optimization of catalysts is desired because we only dealt with exploring the additive oxide of the catalysts. Supported metals and supports instead of Pt and TiO$_2$ should also be explored. For this, we can use the same feature engineering strategy by harnessing the intrinsic properties of supported metals and supports. For instance, we can use "support descriptors" such as specific surface areas, band gaps, and acidity (which can be measured experimentally) for the support materials. In the future, we expect our study to facilitate the development of novel catalysts.

## Methods
### Chemicals

Chemicals and materials were purchased from commercial suppliers and used without further purification. TiO$_2$ (P25) having both anatase and rutile phases was obtained from Evonik (formerly Degussa). TiO$_2$ STR-100N having rutile phase was provided by Sakai Chemical Industry, while TiO$_2$ ST-01 with anatase phase was obtained from Ishihara Sangyo. The carbon and γ-Al$_2$O$_3$ (Puralox) supports were commercially obtained from Kishida Chemical and Sasol, respectively. ZrO$_2$ (JRC-5) was supplied by the Catalysis Society of Japan. SiO$_2$ (CariACT Q-10) was purchased from Fuji Silysia Chemical Company Ltd. Nb$_2$O$_5$ was prepared by calcination of niobic acid (Nb$_2$O$_5 \cdot n$H$_2$O, HY-340) supplied from CBMM (Companhia Brasileira de Metalurgia e Mineração) at 500 °C for 3 h. CeO$_2$ (Type-A) support was provided by Daiichi Kigenso Kagaku Kogyo Co., Ltd. The industrial CuZnAl catalyst known as a copper-based low-temperature water-gas shift catalyst (HiFUEL® W220; CuO = 52 wt%, ZnO = 30 wt%, Al$_2$O$_3$ = 17 wt%) and the FeCrCuO$_x$ catalyst known as an iron–chrome-based high-temperature water-gas shift catalyst (HiFUEL® W210; Fe$_2$O$_3$ = 82.7 wt%, Cr$_2$O$_3$ = 7 wt%, CuO = 5 wt%) were purchased from Alfa Aesar.

### Preparation of the catalysts

Pt(3)/M$_1$(X$_1$)-M$_2$(X$_2$)-M$_3$(X$_3$)-M$_4$(X$_4$)-M$_5$(X$_5$)/TiO$_2$ (3 wt% Pt, TiO$_2$ = P25, X$_x$ is the loading amount of M$_x$) was prepared using the sequential

impregnation method. Elements M having atomic numbers from 3 (Li) to 83 (Bi), except for Be, B, C, N, O, P, S, As, Se, Tc, Te, Pm, Ta, Hg, Tl, halogens, noble gases, and platinum group metals, were used as catalyst components in this work. For the source and purity of the chemicals, please see Supplementary Table 1. First, the single or multiple additive components supported $TiO_2$ ($M_1(X_1)$-$M_2(X_2)$-$M_3(X_3)$-$M_4(X_4)$-$M_5(X_5)$/$TiO_2$) was prepared by the impregnation method. In the process, a mixture of related amount of $TiO_2$ support and corresponding sources of M elements was charged in a 100 mL glass vessel containing an appropriate amount of deionized water and stirred for 15 min with 200 rpm agitation at room temperature. The mixture was evaporated to dryness at 50 °C, dried at 110 °C for 12 h, and calcinated at 500 °C in air for 3 h to give $M_1(X_1)$-$M_2(X_2)$-$M_3(X_3)$-$M_4(X_4)$-$M_5(X_5)$/$TiO_2$. The formed $M_1(X_1)$-$M_2(X_2)$-$M_3(X_3)$-$M_4(X_4)$-$M_5(X_5)$/$TiO_2$ was then impregnated in an aqueous $HNO_3$ solution of $Pt(NH_3)_2(NO_3)_2$ under magnetic stirring. The mixture was evaporated to dryness at 50 °C and further dried in air at 110 °C for 12 h to give $PtO_2$/$M_1(X_1)$-$M_2(X_2)$-$M_3(X_3)$-$M_4(X_4)$-$M_5(X_5)$/$TiO_2$ (unreduced sample). The catalyst used for the RWGS reaction was prepared by reduction of $PtO_2$/$M_1(X_1)$-$M_2(X_2)$-$M_3(X_3)$-$M_4(X_4)$-$M_5(X_5)$/$TiO_2$ in a quartz tube under a flow of $H_2$ (40 mL min$^{-1}$) at 300 °C for 0.5 h to give $Pt(3)$/$M_1(X_1)$-$M_2(X_2)$-$M_3(X_3)$-$M_4(X_4)$-$M_5(X_5)$/$TiO_2$.

Other supported catalysts were prepared by the same method described above by using $M_1(X_1)$-$M_2(X_2)$-$M_3(X_3)$-$M_4(X_4)$-$M_5(X_5)$/$TiO_2$ or $M_1(X_1)$-$M_2(X_2)$-$M_3(X_3)$-$M_4(X_4)$-$M_5(X_5)$/Support and other metal sources including aqueous solutions of $NH_4ReO_4$, $RuCl_3$, $IrCl_3 \cdot nH_2O$, $AgNO_3$ and aqueous $HNO_3$ solutions of $Rh(NO_3)_3$ and $Pd(NH_3)_2(NO_3)_2$.

## Catalysts characterization

High-angle annular dark-field scanning transmission electron microscopy (HAADF-STEM) and energy dispersive X-ray spectroscopy (EDX) analysis were performed using an FEI Titan G2 microscope. Samples were prepared by dropping an ethanol solution containing the catalyst on carbon-supported Cu grids. XPS characterization was carried out on a JEOL JPS-9010MC spectrometer using Mg Kα (1253.6 eV) radiation. Binding energies were calibrated based on the C1s peak energy (285.0 eV). The samples were examined after the $H_2$ reduction pretreatment using a transfer vessel in order to avoid exposure to air. XPS spectra were analyzed by convolution of Gaussian and Lorentzian functions with a Shirley background.

In situ/ *operando* IR spectra were recorded on a JASCO FT/IR-4600 equipped with a mercury-cadmium-telluride detector and a quartz IR cell connected to a conventional flow system (100 mL min$^{-1}$). The sample was pressed into a 40 mg self-supporting wafer and mounted in the quartz IR cell with $CaF_2$ windows. Spectra were acquired by accumulating 20 scans at a resolution of 4 cm$^{-1}$. The reference spectrum of the catalyst wafer in He flow taken at the measurement temperature was subtracted from each spectrum.

Pt L$_3$-edge, Rb K-edge, Mo K-edge, Ba K-edge, and Ti K-edge XAS measurements were performed in a transmission mode, while Nb K-edge XAS were performed in a fluorescence mode at the BL14B2 of SPring-8 at the Japan Synchrotron Radiation Research Institute (Proposal No. 2021B1840 and 2022A1736). A Si(311) double crystal monochromator was used for the Pt L$_3$-edge, Rb K-edge, Nb K-edge, Mo K-edge, and Ba K-edge XAS measurements, while a Si(111) double crystal monochromator was used for the Ti K-edge XAS measurements. For *operando* XAS measurements, a high-sampling-rate TCD GC (490 Micro GC; Agilent Technologies Inc.) was used for the quantitative analysis of CO and CH$_4$. A mass spectrometer (BELMass; MicrotracBEL Corp.) was also used to monitor the eluent gas. Samples in pellet form (∅7 mm) were introduced into a cell equipped with Kapton film windows and gas lines connecting to the GC. Pretreatment of the samples involved heating under a flow of $H_2$ (300 mL min$^{-1}$) at 300 °C for 30 min. Subsequently, 25% $CO_2$/He (400 mL min$^{-1}$), 75% $H_2$/He (400 mL min$^{-1}$), and $CO_2$ (100 mL min$^{-1}$) + $H_2$ (300 mL min$^{-1}$) were

introduced into the cell with intervals of He purge between the gas introduction steps. Note that boron nitride was used to make a pellet sample when the required amount is <40 mg. Spectra of reference compounds were recorded at room temperature in air. The obtained XAS spectra were analyzed using the Athena and Artemis software ver. 0.9.25 included in the Demeter package[54].

## Catalytic reverse water-gas shift reactions

RWGS reactions were carried out in a fixed bed continuous flow reactor under atmospheric pressure. A straight quartz tube with an inner diameter of 4 mm was used. The catalyst (typically 10 mg) was pretreated under $H_2$ flow (40 mL min$^{-1}$) at 300 °C for 30 min prior to each activity test. Catalytic activity was measured at the temperature of 250 °C under the following composition of feed gas: 20 mL min$^{-1}$ $CO_2$, 60 mL min$^{-1}$ $H_2$, and 5 mL min$^{-1}$ $N_2$ added as an internal standard for quantitative analysis. The gas flows were controlled by mass flow controllers. The effluent gas phase was allowed to pass through an ice-bath unit to remove the water vapor and then analyzed online using a gas chromatograph (Agilent 490 Micro GC) equipped with Molsieve 5 Å and PoraPLOT Q columns and TCD detector.

## ML methods

As elemental descriptors, we selected the following eight parameters: electronegativity (EN) according to the Allred-Rochow's definition, melting point (m.p.), enthalpy of formation ($\Delta H_{fus}$), density, the group of the periodic table, BG in the most stable oxide from, oxidation number in the most stable oxide form, and adsorption energy ($E_{ads}$) of $CO_2$ on the metallic surface.

We used ETR[38] as an ML model. Widely used implementations of scikit-learn (version 0.23.2)[55] were employed for all ML models. For hyperparameter tuning, we tested a reasonable range of candidate values in an exhaustive way (grid search) shown in Supplementary Table 2, chose the best hyperparameter by 5-fold CV on the training set, and used the model for calculating the predicted values for the test set (the hyperparameters not explicitly indicated in the table were set to the scikit-learn defaults). Namely, to avoid data leakage, we strictly followed a standard practice of "nested" CV, also known as double CV, to estimate the prediction accuracies; we used 5-fold CV for the internal CV, and used Monte Carlo CV (also known as repeated random subsampling CV) with 100-times of random leave-20%-out trials for the external CV to increase the statistical reliability for validating the test prediction accuracies with fixing the number of training data.

We have used three types of ML approaches that differ in the input representations of catalysts; (i) naive ML model that uses only elemental compositions, (ii) exploitative ML model that uses both elemental compositions and elemental properties, and (iii) *explorative* ML model that uses only elemental properties. For the input representations of elemental compositions, each catalyst was represented as a vector of compositional fractions of each element for all 50 elements under consideration, i.e., $(c_1, c_2, c_3, \cdots, c_{50})$ where $c_i$ is the compositional fraction of the i-th element. For the input representation of elemental properties, each catalyst is represented as the sum of vectors of each elemental descriptor scaled by its compositional fraction, i.e., for a catalyst $Pt(3)$/$M_1(X_1)$-$M_2(X_2)$-$M_3(X_3)$-$M_4(X_4)$-$M_5(X_5)$/$TiO_2$,

$$X_1\,vec(M_1) + X_2\,vec(M_2) + X_3\,vec(M_3) + X_4\,vec(M_4) + X_5\,vec(M_5), \quad (1)$$

where $vec(M_i)$ is the elemental descriptor vector for element $M_i$, which is also called the composition-based feature vector in the literature[33]. The former representation generates 50-dimensional features and tends to be very sparse and statistically uninformative when the training dataset is not large but contains many elements. Moreover, it is incapable of handling elements that are absent or statistically

infrequent in the training data. On the other hand, the latter representation has the same dimension as the user-specified elemental descriptor that often produces statistically much more stable results for small-data problems and is not explicitly constrained by the elements covered in the training dataset. Moreover, technically, in the latter representation, each catalyst is represented as a set of elemental descriptors and scaled by its composition fraction and aggregated into a single feature vector for the given catalyst by sum pooling, a permutation-invariant operation.

Notably, the explorative ML model that represents catalysts only with respect to their physico-chemical properties via certain descriptors without directly specifying the individual contributions of distinct elements, enables a more extrapolative and ambitious exploration beyond the training data even to find unseen elements. In our previous study utilizing these ML approaches for the analysis of reaction data on oxidative coupling of methane (OCM)[31], we also developed a procedure to recover the catalyst composition from the elemental property representation because the composition information is indispensable for catalyst synthesis. We employed a "local search" to find new catalyst candidates. However, in the present study, we employed the "grid search" approach to suggest new catalyst candidates by manually specifying the loading amount of each element M in order to perform global optimization. In this approach, we do not need to use the recovery procedure but rather calculate the expected improvement (EI)[56] score that is obtained using the following equation for the given compositions.

$$EI(x) = \mathbb{E}\{\max(\mu(x) - y^*, 0)\} = (\mu(x) - y^*) \cdot \Phi\left(\frac{\mu(x) - y^*}{\sigma(x)}\right) \\ + \sigma(x) \cdot \phi\left(\frac{\mu(x) - y^*}{\sigma(x)}\right) \quad (2)$$

Here, $\mu(x)$ and $\sigma(x)$ are the predicted value and the standard deviation of an ML surrogate for an input $x$, while the expectation $\mathbb{E}$ assumes a Gaussian distribution with a PDF of $\phi$ and CDF of $\Phi$. EI scores can be intuitively considered as a quantity that indicates how much improvement over the current best $y^*$ can be expected for an input $x$. The EI is schematically presented in Supplementary Fig. 3.

Clustering was typically performed to group very similar candidates into K clusters. In cases where clustering was not used, we simply selected the catalysts based on the top proposed catalyst compositions. We normally used $K = 100$ because the elbow and silhouette analyses suggested that 100 was the optimal number of clusters. The elbow method was employed to find the point of inflection (elbow) in the plot of the explained variation as a function of the number of clusters, serving as a criterion for determining the optimal number of clusters. The silhouette analysis was applied to quantify the similarity among the observations within a cluster, thus providing additional support for identifying the optimal number of clusters. A representative analysis result using the 300 data points (See the data directory in the GitHub repository https://github.com/shinya-mine) with explorative ML methods based on ETR (Supplementary Fig. 5) revealed that $K = 50$–$100$ is optimal. In addition, no clusters had silhouette scores below the average when $K = 100$ (with $N = 10$ perturbations).

### Procedure of ML-assisted RWGS catalysts discovery
The initial dataset consisting of 45 data points was constructed using catalysts reported in our previous experimental study and some new catalysts synthesized for the present study, as given in the data directory of our GitHub repository and labeled as "Iteration" = 0 (https://github.com/shinya-mine). We suggested the next catalyst candidates using the explorative ML model based on ETR and the initial dataset (45 data points), picked some suggested catalysts according to the EI ranking, synthesized the catalysts using the

sequential impregnation method, performed the RWGS reaction, and updated the dataset to close the loop (Supplementary Fig. 1). Subsequently, we suggested the next catalyst candidates using the explorative ML model based on ETR and the updated dataset (50 data points) and performed the experiments according to the ML prediction to further update the dataset. We continued this procedure until we performed 44 loops to test 300 catalysts. Since we typically performed the clustering with $K = 100$, as mentioned above, our ML pipeline gave a list of 100 top-ranking candidates at each iteration, and we chose the catalysts for the actual experiments from this list. As it is practically difficult to test all the 100 candidates in actual experiments, only some of the suggested catalysts were tested (i.e., not all the 100 candidates were experimentally tested). The selection from the top 100 candidates suggested by the ML approach was manually performed by considering the diversity of the catalyst compositions. ETR was used throughout in this study. Only the explorative ML model was used for the initial effort because we wanted to explore many elements and its prediction accuracy was the highest among the three ML models at the initial stage while the exploitative ML model was also used after 30 iterations.

## Data availability
The source data, which support the result of this study, can be found in the manuscript and Supplementary information. All experimental data used for machine learning are available in Excel format on the URL and can be freely used (https://github.com/shinya-mine).[57]

## Code availability
All machine learning codes used in this study were written under the anaconda distribution environment of python3 (https://www.anaconda.com) and can be found online at https://github.com/shinya-mine[57]. The VASP code package used in this work can be accessible after a user license is authorized by the VASP company (https://www.vasp.at).

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

## Acknowledgements

This study was financially supported by the KAKENHI grants (21K18185 (T.T. & I.T.) and 22K14538 (S.M.)) from the Japan Society for the Promotion of Science (JSPS), JST-CREST Program JPMJCR17J3, JST-FOREST Program JPMJFR211U, and the Joint Usage/Research Center for Catalysis. G.W. acknowledges the JSPS postdoctoral fellowship (P20345) (G.W.). A portion of the calculations was performed on supercomputers at RIIT (Kyushu University) and ACCMS (Kyoto University).

## Author contributions

G.W., D.C., S.M., and T.Y. synthesized the catalysts and performed the catalytic reactions. S.M., M.T., and I.T. composed the ML codes and conducted ML predictions and analysis. S.M., G.W., D.C., Y.J., and K.W.T. characterized the catalysts. S.M., G.W., Y.J., and Z.M. conducted the operando spectroscopic experiments. K.M. provided insights into the experimental work. I.T., K.S., and T.T. directed the project and provided guidance for the experimental and computational work. The manuscript was written by S.M., G.W., D.C., I.T., and T.T. with input from all authors.

## Competing interests

The authors declare no competing interests.
