## [Peer Review File · Nature Communications]

Accelerated discovery of multi-elemental reverse water-gas shift catalysts using extrapolative machine learning approachEditorial Note: This manuscript has been previously reviewed at another journal that is not operating a transparent peer review scheme. This document only contains reviewer comments and rebuttal letters for versions considered at *Nature Communications*.

REVIEWER COMMENTS

Reviewer #1 (Remarks to the Author):

The details regarding the ML approach are still generally confusing, e.g., clustering was used to select the catalysts? Was this always done and just not mentioned previously?

Additionally, the authors claim that the search space was approximately 800 billion and that roughly six catalysts were selected based on the expected improvement from that large possible space.

The impression that the authors give is that ~6 catalysts were selected from 800 billion based on the expected gain at each iteration. However, I am uncertain whether the three different ML models used converged on the same set of 300 catalysts... If that's what wasn't done, then what is the point of training these three different ML models at each step?

Moreover, the authors have not provided the actual predictions of the expected gain/improvement (EG) for each of the predictions at each iteration. Yes, the expected gain is not directly measurable experimentally but it should still be correlated with the measured CO-formation rate.

The authors have also sidestepped important concerns raised by reviewers regarding the overfitting of their model. Although the overfitting is labeled as “benign” by the authors, it may explain why the model still predicts samples over the range of 2.0 - 4.0 mmol min⁻¹ gcat⁻¹ (i.e., 50% of the total range of known samples) at the end of 44 loops.

On this topic, how is the R² calculated in Fig. 1? From a separate model that was CO-formation or based on the EG with the CO formation rate? Were the R² values were calculated on the 10% test set of the model at that iteration before experimental validation or on the ~6 compounds that were subsequently experimentally validated?

The caption for Fig. 1 still does not explain what the grey dots are in the bottom left panel, and I am struggling to understand the takeaway from Fig. 3, in particular why several different approaches for feature ranking and what the importance points are. Moreover, it's unclear what ML model (and the target property it was trained on) the SHAP analysis in Fig. 3 was performed on.

Overall, the paper is selling a type of experiment-ML loop that is too contrived and not adaptable for general use. The paper lacks clarity, and I still have several questions and concerns about the figures and the

methods used in the paper. I cannot recommend publication until the authors address these issues. More generally, based on the content of this paper, I believe it would be more appropriate for submission to a journal that has a specific focus on catalysis.

Reply to the comments of referee 1

Comment 1:

The details regarding the ML approach are still generally confusing, e.g., clustering was used to select the catalysts? Was this always done and just not mentioned previously?

Reply to comment 1:

Our ML pipeline gives a list of 100 top-ranking candidates at each iteration, and we choose the catalysts for actual experiments from this list. When this list includes similar catalysts, we perform clustering. We do not test any catalysts that were not included in the suggested list, and in that sense, our experimental design is driven by our ML approach. However, as it is practically difficult to test all of the 100 candidates in actual experiments, and thus the selection from the top 100 candidates by ML are manually done (Not all of 100 were experimentally tested). Indeed, we may have missed some potentially better performing candidates in this manual process. But even with that imprecision, we have succeeded in identifying a superior catalyst via the ML approach proposed in this study. To clarify the aforementioned information, we have included the following text in the revised manuscript.

(Page 19)

“Clustering was typically performed to group very similar candidates into K clusters. In cases where clustering was not used, we simply selected the catalysts based on the top proposed catalyst compositions.”

Comment 2:

Additionally, the authors claim that the search space was approximately 800 billion and that roughly six catalysts were selected based on the expected improvement from that large possible space.

The impression that the authors give is that ~6 catalysts were selected from 800 billion based on the expected gain at each iteration. However, I am uncertain whether the three different ML models used converged on the same set of 300 catalysts... If that's what wasn't done, then what is the point of training these three different ML models at each step?

Reply to comment 2:

The three different ML models described in the manuscript would achieve convergence of different sets of 300 catalysts. The discovery process needs to consider two aspects, exploitation and exploration, and needs to take a balance between them. We need to acquire new information about unexplored areas and make use of obtained information from already explored areas. Our basic strategy in this study was to use the explorative method until we obtained sufficient information. As mentioned in the main text, our study mainly used the explorative ML model and only used the exploitative ML model after the prediction accuracy reached a certain level (after 30 iterations). The naive ML model was not used for the catalyst discovery process, but we ensured to provide the prediction results obtained with such unused models for comparison because favorable featurization techniques are of great interest to the readers. In addition, fractional representation in a one-hot encoding manner is known to perform as well as or better than many other featurization techniques when large datasets are used. We believe that the size of our dataset is typical for applying ML to many challenges in solid catalysis, in particular, and chemical and materials science fields, in general. Therefore, we consider it important to present the prediction accuracy of each featurization method in our study. The following sentences have been modified for clarity:

(Page 3)

“The explorative ML model was used in the initial effort to explore many elements, and because the model achieved the highest prediction accuracy among the three ML models. The exploitative ML model was used after the prediction accuracy reached a certain level (after 30 iterations). Although the naive ML model was not used for the catalyst discovery process in this study, its prediction results are given for comparison, because fractional representation in a one-hot encoding manner is known to perform as well as or better than many other featurization techniques when large datasets are used²⁹.”

Comment 3:

Moreover, the authors have not provided the actual predictions of the expected gain/improvement (EG) for each of the predictions at each iteration. Yes, the expected gain is not directly measurable experimentally

but it should still be correlated with the measured CO-formation rate.

Reply to comment 3:

We highly appreciate the referee's comment, but we respectfully think that direct comparison between the expected improvement scores and the actual improvement amounts is invalid. In brief, the expected improvement scores are expectations with respect to the uncertainty of the ML model, not the uncertainty of the target phenomenon itself. The EI scores can also be large when the ML model is quite uncertain about a catalyst candidate simply owing to its lack of similar data at the point of prediction. This holds true not only for our method but also for other standard methods.

This point is demonstrated in the following typical illustrative example using the most widely used method of Bayesian optimization with Gaussian process:

https://scikit-optimize.github.io/stable/auto_examples/bayesian-optimization.html.

The objective is to find the value of x that minimizes the unknown function f by repeatedly selecting and evaluating values for x . As shown in Figure R1 (a), we begin with five randomly selected initial points. It is important to note that the observations of $f(x)$ are always accompanied by 1% random noise and that the true values of $f(x)$ are never observed directly. Figure R1 (b) shows that this method successfully identifies a point x that gives the (nearly) minimum value of $f(x)$ after only a few iterations. We can observe that the current best values decrease within only four iterations. However, as illustrated in Figure R1 (c), there is no correlation between the expected improvement (EI) scores and the actual improvements, despite the overall success of our procedure. Out of the ten tests conducted, only three cases resulted in actual improvement. Figure R2 provides detailed plots of the EI scores, ML model predictions, ML model uncertainty, and true values at each iteration. At iteration '0', we fit the ML model to the five initial points (the ML prediction is shown as a green dashed line) and evaluate its prediction uncertainty (in light green). It is important to note that this problem involves a small sample size, resulting in significant uncertainty in ML predictions for most areas. However, the uncertainty decreases as more data points are collected. Then, we compute the EI scores from the ML model trained on the obtained data points and plot them above each box. We select the point x_{next} with the highest EI score for experiment '1'. This approach worked for the '0' case, resulting in actual improvements. Meanwhile, in the '2' case, we selected a point x_{next} with a very high EI score, but the actual value of $f(x_{next})$ was found to be quite large, leading to negative improvement if it were to be used (i.e., discarding it would yield better results than using it). This discrepancy occurs because of the lack of informative data around x_{next} , and the significant uncertainty that ML prediction carries. The high EI score reflects the potential for improvement simply because we have not yet explored it, despite the true $f(x_{next})$ value not being small. In such cases, the EI scores are high because testing x_{next} for exploration rather than exploitation is necessary to gather new information. Therefore, the ML discovery process strikes a balance between exploration and exploitation. Moreover, it becomes challenging to find further improvement after the fourth iteration, where we have already obtained the nearly minimum value. In this situation, the EI scores also lean towards more exploration, resulting in the exploration of both ends and several already-explored points in between.

Fig. 1 in the main manuscript shows that our ML approach worked as expected and produced multiple nontrivial high-performance catalysts. For information, we also show the comparison between the EI scores and the amounts of actual improvement in Fig. R3, with the caution that such comparison is uncommon not only for our method but also for other standard methods where the EI scores are widely used such as Bayesian optimization and kriging.

Figure R1. (a) A toy example problem of sequential model-based optimization. The goal is to find the point x that results in the minimum $f(x)$ by iteratively selecting new points to evaluate. (b) The plot of current best $f(x)$ values found at each iteration by Bayesian optimization. The procedure successfully identified the point x giving the nearly minimum value of $f(x)$ after 4 iterations. (c) The comparison between EI (expected improvements) scores of the model and the actual amounts of improvement, showing no clear correlations despite achieving successful minimization.

Figure R2. The plots of $EI(x)$ (top panels), $f(x)$ (light red line), ML prediction of $f(x)$ (green dashed line), ML uncertainty for $f(x)$ (light green) at each i -th iteration. For example, '0' indicates the 0-th iteration where we only fit an ML model (Gaussian process regressor) to the initial points and calculate the query point x_{next} to evaluate that is the point x giving the largest $EI(x)$. If the true $f(x_{next})$ value evaluated for the next iteration is smaller than the current best (minimum) values among the already-observed $f(x)$ values, we label it as "improved."

Figure R3. (a) Actual improvement (observed CO formation rate - best CO formation rate obtained up to each iteration) for each catalyst. (b) Correlation of expected improvement (EI) and actual improvement.

Comment 4:

The authors have also sidestepped important concerns raised by reviewers regarding the overfitting of their model. Although the overfitting is labeled as "benign" by the authors, it may explain why the model still predicts samples over the range of 2.0 - 4.0 $\text{mmol min}^{-1} \text{g}_{\text{cat}}^{-1}$ (i.e., 50% of the total range of known samples) at the end of 44 loops.

Reply to comment 4:

The CO formation rate in Fig 1 is the result of experimental validation. As we explained in the previous revision round, our dataset is still relatively small compared with the search space. Therefore, we have not yet achieved high accuracy in experimental validation to collect new data points beyond our current dataset, even after performing the 44 loops. However, the proposed study design of this article is an iterative one, i.e., a sequential experimental design. Thus, the focus is placed more on how to make use of the available data (even if the dataset is small in the statistical sense) to plan subsequent experiments towards better catalyst discovery.

In addition, the concern raised by the referee (the model still predicts catalysts over the range of 2.0 - 4.0 mmol min⁻¹ gcat⁻¹ at the end of 44 loops) can technically occur during the exploration, such as the case after iteration 4 in Fig. R1, in which we already have good values as the current best to compute the EI. Further, and unlike the standard ML setting where training and testing datasets come from the same distribution, our discovery setup is largely explorative with the simultaneous extension of the small collectible dataset. In this situation with only a small dataset to use, the ML prediction itself might not work, because ML model is merely a representative of the available data.

We performed a standard procedure (double CV) to evaluate our model and found that ETR produced the best prediction accuracy among the six ML models used in this study. Therefore, we believe that our ML method based on ETR serves as the best model to explore new catalysts. To highlight this fact, the following sentences have been added to the revised manuscript.

(Page 4)

“Although our dataset is still small (300 data points) and the best prediction accuracy we attained after 44 cycles ($R^2 = 0.81$) is not significantly high, the proposed design is iterative, i.e., a sequential experimental design. Thus, the focus is placed more on how to utilize the available data (even if the dataset is small in the statistical sense) to plan subsequent experiments and achieve better catalyst discovery. We believe that the prediction accuracies (up to $R^2 = 0.81$) achieved by a standard cross validation (CV) procedure (see the ML methods section for details) would be sufficient to statistically sense promising directions for further research. It is also noteworthy that the obtained prediction accuracy ($R^2 = 0.81$) is somewhat higher than those attained in most ML studies using experimental data on heterogeneous catalysis and relevant material science topics, wherein the prediction accuracy is typically below $R^2 = 0.8$, even when experimental conditions are used as descriptors^{28,31,32,39–42}.”

Comment 5:

On this topic, how is the R^2 calculated in Fig. 1? From a separate model that was CO-formation or based on the EG with the CO formation rate? Were the R^2 values were calculated on the 10% test set of the model at that iteration before experimental validation or on the ~6 compounds that were subsequently experimentally validated?

Reply to comment 5:

The R^2 values were calculated using the cross validation (CV) method described in the “ML methods” section on the dataset at each iteration before experimental validation. The caption of Fig. 1 has been modified as shown below to clearly convey this information.

Fig. 1. ML-assisted exploration of RWGS catalysts. (A) ML-assisted exploration of RWGS catalysts using the explorative and exploitative ML methods based on ETR. Catalysts with elements not seen in the original dataset are shown with diamond-shaped markers while catalysts with elements seen in the original dataset are shown with grey color and circle-shaped markers. The solid red line shows the best CO formation rate at each iteration, and for comparison, the dashed navy and dash-dotted green lines show the CO formation rates for Pt(3)/Mo(10)/TiO₂ and Cu/ZnO/Al₂O₃ catalysts, respectively. The R² values were calculated using the cross validation (CV) method described in the ML methods section on the dataset at each iteration before experimental validation. (B) Radar charts of the elemental descriptors for the best catalysts at each iteration. Descriptor values relative to the (i) Pt(3)/Mo(10)/TiO₂ catalyst are shown.

Comment 6:

The caption for Fig. 1 still does not explain what the grey dots are in the bottom left panel, and I am struggling to understand the takeaway from Fig. 3, in particular, why several different approaches for feature ranking and what the importance points are. Moreover, it's unclear what ML model (and the target property it was trained on) the SHAP analysis in Fig. 3 was performed on.

Reply to comment 6:

The caption for Fig. 1 has been modified as indicated above. Regarding Fig. 3, we have provided the additional explanation shown below. We used several methods for interpreting the ML models because different “interpretable ML” methods can produce different results owing to slightly different assumptions in each model. Therefore, it is generally preferable to use multiple models/methods to focus on any common trends across the models/methods. Regarding the latter point, we trained the ML model to “predict CO formation rates of all the 300 catalysts in our final dataset” as explained in the original caption of Fig.3.

(Page 8)

“The waterfall plot analysis reveals the descriptors that are responsible for the increase or decrease from the average value of the dataset (2.28) relative to the predicted value for each catalyst.

EN, group, and oxide band gap (BG) values were found to strongly contribute to the high activity of our best catalyst (Pt(3)/Rb(1)-Ba(1)-Mo(0.6)-Nb(0.2)/TiO₂). Note that the summary plot shown in Fig. 3B gives explanations for overall predictions for all the datapoints used (300 datapoints here) whereas the waterfall plots (Fig. 3C,D) are designed to display explanations for individual predictions for each catalyst^{48,49}. This difference in methodology is reflected in the differences in the ranking of important descriptors in each analysis method. Therefore, the summary plot is useful for obtaining information on the catalyst design guidelines for the RWGS reaction in general, whereas the waterfall plots provide more useful information on the reasons behind the high (or low) activity shown by an individual catalyst.”

Comment 7:

Overall, the paper is selling a type of experiment-ML loop that is too contrived and not adaptable for general use. The paper lacks clarity, and I still have several questions and concerns about the figures and the methods used in the paper. I cannot recommend publication until the authors address these issues. More generally, based on the content of this paper, I believe it would be more appropriate for submission to a journal that has a specific focus on catalysis.

Reply to comment 7:

To address the referee’s comments, we have further revised the manuscript for clarity. In addition, the data and code used in our research have been made available on our GitHub page for those wishing to use them to conduct their own work.

An increasing number of studies apply AI to search for heterogeneous catalysts. However, most of these studies work on well-behaved computational data, and studies that experimentally validate their findings are still rare. In addition, even in ML studies using experimental data, the findings typically only confirm already known high-performance catalysts and/or do not provide novel catalysts that show higher performance than those in their original datasets. Our work presents a new approach to discovering novel catalysts and materials that exhibit extraordinary performance. In addition, this study overcomes the most common limitation and criticism of ML, that is, its inability to extrapolate. More specifically, although conventional ML models can only work with frequently appearing elements in the dataset, our ML method, which considers elemental properties, was able to find originally unseen elements. The discovered Pt(3)/Rb(1)-Ba(1)-Mo(0.6)-Nb(0.2)/TiO₂ catalyst contains Nb, which was not included in the original dataset. This catalyst was experimentally confirmed to show state-of-the-art activity. Such unexpected and surprising results are rarely available from either the field of catalysis or the broader field of materials science. The concept of our research methodology for finding truly innovative catalysts is not only relevant to thermal heterogeneous catalysis, but also more generally applicable to other challenges in catalysis, chemistry, and materials science. For example, batteries, electronic materials, fluorescent materials, magnetic materials, etc., are typically composed of complex multi-component systems for which the demand for advanced designs continues to grow. Furthermore, high entropy alloys/oxides, which have recently garnered attention across various disciplines of material science, can also be designed using basically the same approach. Therefore, we believe that our study will be of high interest to the broad readership of Nature Communications.

REVIEWERS' COMMENTS

Reviewer #1 (Remarks to the Author):

My questions were addressed, and explanations were provided where I had highlighted missing information.

The only remaining concern I have pertains to reconciling this sentence in the Author's reply with the claims made in the manuscript:

"However, as it is practically difficult to test all of the 100 candidates in actual experiments, and thus the selection from the top 100 candidates by ML are manually done (Not all of 100 were experimentally tested)."

I interpret this sentence as suggesting that the authors did not solely rely on the ML model to select the catalyst with the highest EI in each iteration. Instead, they employed their domain expertise to synthesize catalysts from a pool of the top 100 candidates. Therefore, the selection process could be considered a combination of both manual selection and ML model-driven selection, which should be mentioned.

Response to the comments of reviewer #1

Comment:

My questions were addressed, and explanations were provided where I had highlighted missing information. The only remaining concern I have pertains to reconciling this sentence in the Author's reply with the claims made in the manuscript:

"However, as it is practically difficult to test all of the 100 candidates in actual experiments, and thus the selection from the top 100 candidates by ML are manually done (Not all of 100 were experimentally tested)."

I interpret this sentence as suggesting that the authors did not solely rely on the ML model to select the catalyst with the highest EI in each iteration. Instead, they employed their domain expertise to synthesize catalysts from a pool of the top 100 candidates. Therefore, the selection process could be considered a combination of both manual selection and ML model-driven selection, which should be mentioned.

Response:

We thank you very much for the endorsement for publication and for your helpful suggestion. Based on your suggestion, the following text has been added to the main manuscript.

(Page 14)

Since we typically performed the clustering with $K = 100$, as mentioned above, our ML pipeline gave a list of 100 top-ranking candidates at each iteration, and we chose the catalysts for the actual experiments from this list. As it is practically difficult to test all the 100 candidates in actual experiments, only some of the suggested catalysts were tested (i.e., not all the 100 candidates were experimentally tested). The selection from the top 100 candidates suggested by the ML approach was manually performed by considering the diversity of the catalyst compositions.